# Effects of Adenotonsillectomy on Vocal Function

**DOI:** 10.3390/jpm13061002

**Published:** 2023-06-15

**Authors:** Massimo Mesolella, Giovanni Motta, Salvatore Allosso, Gaetano Motta

**Affiliations:** 1Unit of Otorhinolaryngology, Department of Neuroscience, Reproductive Sciences and Dentistry, University Federico II of Naples, 80131 Naples, Italy; 2Unit of Otorhinolaryngology, University Luigi Vanvitelli, 80131 Naples, Italy; giovannimotta95@yahoo.com (G.M.); gaetano.motta@unicampania.it (G.M.)

**Keywords:** adenotonsillectomy, mouth breathing, dysphonia, vocal function, respiratory dynamics

## Abstract

Introduction: Correct breathing is a fundamental condition for adequate vocal production. Respiratory dynamics are able to modify the growth of facial mass and lingual posture, i.e., of the skull, the mandibular one. For this reason, infant mouth breathing can cause hoarseness. Materials and methods: We evaluated the actual changes in the characteristics of the voice and articulation of language in a group of subjects affected by adenotonsillar hypertrophy (grade 3–4), with frequent episodes of pharyngo-tonsillitis who underwent adenotonsillectomy. Our study included 20 children—10 boys and 10 girls—aged 4 to 11 years who had adenotonsillar hypertrophy and pharyngotonsillitis episodes exceeding 5–6 per year in the previous 2 years. The control group (Group B) included 20 children—10 boys and 10 girls—aged 4 to 11 years (average age of 6.4 years) who had not undergone surgery and shared the same degree of adenotonsillar hypertrophy as those in Group A but who did not experience recurrent pharyngotonsillitis episodes. Discussion: The hypertrophy of adenoids and tonsils significantly impacted breathing, vocal function, and speech articulation. All this is responsible for a state of tension in the neck muscles, which at the level of the vocal tract causes hoarseness. The changes objectively observed in our study in the pre- and post-operative phase demonstrate how adenotonsillar hypertrophy is responsible for an increase in resistance to the passage of air at the glottic level. Conclusions: For this reason, adenotonsillectomy has an impact on recurrent infections and can also lead to an improvement in speech, breathing, and posture.

## 1. Introduction

Optimal breathing is a fundamental requirement for adequate vocal production. Thoracic-abdominal breathing provides sufficient air for all vocal demands by moving the ribs and lowering the diaphragm. Thoracic breathing is mostly accompanied by oral breathing. This can be habitual or mandatory. In the former case, the individual prefers the oral route for breathing even without nasal obstruction (e.g., bad habits, posture defects). In the latter case, the patient cannot breathe through the nose due to partial or complete and constant obstruction of the upper respiratory tract (adenotonsillar hypertrophy, septal deviations, inferior turbinate hypertrophy, etc.) and must therefore breathe through the mouth [1].

The dynamics of respiration can affect the growth of facial bones, the position of the tongue, the skull, and the mandible, among other structures. As in mandatory oral breathing, a change in the position of the mandible and tongue necessitates a change in the posture of the cervical spine, which is connected to it via the hyoid bone. Furthermore, the position of the mandible impacts the posture of the head, scapula, and pelvic girdle [1,2].

Oral breathing: The position of the mandible, with the mouth open in posterior rotation, causes an imbalance between the forward and upright positions with thoracic kyphosis. This results in hypertonicity of the posterior neck muscles, which prevents the head from falling forward, and hyperkinesia of the anterior muscle districts, which acts as a counterbalance.

The process of inspiration relies on the neck muscles, which tend to move forward; the upper portion of the back arches and the scapulae are raised with each breath. This action helps to expand the thoracic cage. However, this can aggravate the imbalance between the flexor and extensor muscles of the trunk.

During inspiration, the diaphragm is maintained in a raised position while the larynx is retracted. The vocal tract is elongated to raise the pitch, and the vocal frequency increases, leading to a glottic effort with a harsh vocal attack, decreased amplitude and duration of the pre-phonatory and phonatory phases, compensatory action of the false vocal cords, and relative dysphonia [1,2,3].

Based on the information provided, it can be inferred that upward and downward laryngeal stretching can lead to laryngeal muscle stress and, consequently, dysphonia [1,2,3].

### Objectives of the Study

This study was conducted to evaluate the actual changes in voice characteristics and language articulation among a cohort of patients with grade 3–4 adenotonsillar hypertrophy and frequent pharyngotonsillitis-related inflammatory episodes (>5/6 episodes per year) who underwent adenotonsillectomy (Group A). The study evaluated the pre- and post-operative clinical, speech, and instrumental parameters and compared them with a control group (Group B) of patients with similar degrees of adenotonsillar hypertrophy but without recurrent pharyngotonsillitis-related inflammatory episodes.

Thus, in this study, the prevalence of voice, language, and breathing disorders in Groups A and B were evaluated. Moreover, the variations in psychoperceptual characteristics of voice and sound quality in patients who underwent adenotonsillectomy (Group A) were also evaluated. Additionally, any alterations in the language in Group A following surgery were evaluated, and any changes in aerophonic assessment data at the pre- and post-operative stages were evaluated in Group A.

## 2. Materials and Methods

### 2.1. Sample Collection

Our study included 20 children, 10 boys and 10 girls, aged 4 to 11 years (average age of 6.8 years), who had adenotonsillar hypertrophy and pharyngotonsillitis episodes exceeding 5–6 per year in the previous 2 years. The patients were treated from Federico II University Hospital in Naples during 2022. Before undergoing adenotonsillectomy (Group A), all patients underwent a comprehensive evaluation of personal and family medical history, pediatric examination, along with cardiology and anesthesiology assessments [4,5,6,7].

The control group (Group B) included 20 children, 10 boys and 10 girls, aged 4 to 11 years (average age of 6.4 years), who had not undergone surgery and shared the same degree of adenotonsillar hypertrophy as those in Group A, but who did not experience recurrent pharyngotonsillitis episodes.

The selected patients were evaluated by a multidisciplinary team consisting of an otorhinolaryngologist and a speech therapist. The investigation protocol (Appendix A) was also designed based on previously published clinical trials examining the effects of adenotonsillar hypertrophy on the vocal spectrum and language articulation [5,6,7,8,9,10,11,12,13,14,15,16,17,18,19,20,21,22].

Therefore, the patients were evaluated on the following parameters before and three months after adenotonsillectomy surgery:(1)Medical History included information such as the following:-the mode of respiration as reported by parents (oral, nasal, or mixed);-the presence of breathing difficulties, nocturnal snoring, and/or obstructive apnea;-the presence of allergies; and-resting oral posture (closed, partially closed, or open).


This initial data was obtained through interviews with parents and direct observation of patients. Doctors were required to comprehend the subject’s respiratory habits and preferred respiratory pathways. Furthermore, evaluating breathing difficulties, obstructive apnea, and allergies can help diagnose and treat sleep disorders or breathing problems during sleep. The oral posture at rest can also provide valuable information in this regard, such as:-the presence of closed nasality; and-the presence of subjective symptoms associated with phonatory activity (e.g., burning sensation, need to clear the throat, vocal fatigue, unstable voice, the feeling of a foreign body, muscular pain, etc.).
(2)Otorhinolaryngological Clinical Investigation:

To objectively evaluate the clinical condition of the patient, an otorhinolaryngological clinical investigation was conducted. Particularly, for the study of adenotonsillar hypertrophy, Mackenzie’s classification into four grades was taken into consideration:-Grade 1 was characterized by intraglandular tonsils located behind the anterior tonsillar pillar.-Grade 2 tonsils were slightly hypertrophied and protruded just beyond the anterior pillar.-Grade 3 tonsils were hypertrophic and occupied three-quarters of the available space in the throat.-Grade 4 was characterized by hypertrophic tonsils that completely obstructed the airway, causing them to sometimes touch each other (known as “kissing tonsils”) [5,6,7,8,9].
(3)Logopedic Evaluation of Pneumophonic Function:

The logopedic evaluation of pneumophonic function was conducted via the emission of automatic series (e.g., days of the week, months of the year, numerical series, etc.) and spontaneous speech to define the psychoperceptual parameters of voice quality (pitch, intensity, vocal timbre, phonation onset, phonatory stability, and duration of phonation), and the respiratory dynamics in phonation.

(4)Speech and Language Assessment:

The patient’s speech and language were evaluated by administering a phonetic inventory. The inventory required the repetition of phonemes and words and the naming of a series of images containing the target phonemes, with increasing levels of complexity. This examination enabled us to determine the percentage of:-Stable phonemes refer to the phonemes a child has acquired and uses reliably and systematically in his linguistic repertoire.-Occasional phonemes refer to the sounds that are consistently present in a child’s language but are occasionally replaced by other articulatory points.-Absent phonemes refer to articulating words absent in the child’s language.

This analysis enabled us to identify the presence of system simplifications involving the substitution of one phoneme with another and structural simplifications involving modifications to the syllabic structure of the word.

(5)Objective Aerophonic Examination:

The data was collected using an aerodynamic recording system called Aerophone II (F.J.Electronics, Ellebuen 21, DK- 2950. Vedbaek, Denmark). This instrument consists of a computerized system with three transducers that enables the simultaneous measurement of phonatory flow (l/s), sound intensity (dB SPL), and intraoral pressure (cm H_2_O). The three transducers are connected to a facial mask that adheres to the subject’s face during the phonation examination.

The aerodynamic examination consisted of several tests, of which only the Articulatory Test (AT) focused on producing bilabial plosive v-c-v phonemes like I-PI-PI. A silicone tube was inserted into the oral cavity to measure the endoral pressure during this test. The results of this test were analyzed to determine the Phonatory Flow Rate (PFR) and the mean peak endoral pressure during the plosive phase of the/p/sound, which corresponds to the value of Subglottic Pressure (Psub). Thus, we considered the following clinically relevant indices:-the phonatory flow rate index (Phonatory Flow Rate, phonatory flow rate (PFR) index in l/s) represents the volume of airflow per unit of time during vocalization; and-the subglottic pressure (cm H_2_O) expresses the relationship between the pneumo-dynamic activity and the glottis in the phonatory position [7,8,9,10].

### 2.2. Statistical Analysis

A univariate descriptive analysis of the variables under study was performed for the treatment group (Group A—pre surgery) by calculating centrality and variability indices for the quantitative variables and frequency tables for the qualitative variables. The characteristics of this group were then compared with those of the control group (Group B) by the Chi-squared test for the qualitative variables and by the Mann-Whitney non-parametric test for independent samples for the quantitative ones. The use of the non-parametric test was chosen after testing the distributive normality of the quantitative parameters using the Shapiro-Wilks test. Finally, the parameters of interest between pre-surgery Group A and post-surgery Group A were compared using Chi-square for qualitative variables and Wilcoxon’s non-parametric paired-samples test for quantitative variables. An alpha significance level of 0.05 was used in all of the analyzes cited. IBM SPSS Statistics version 28 software was used for the statistical analysis of the data.

The study was approved by the institutional review board committee of Federico II University of Naples, Naples, Italy (2021/208742). Informed written consent from the patients was obtained.

## 3. Results

This study helped us comprehensively overview the main changes in vocal function, language articulation, and respiration after adenotonsillectomy.

During the pre-operative evaluation, all 20 patients in Group A (Table 1), i.e., those with adenotonsillar hypertrophy who underwent adenotonsillectomy, complained of respiratory problems.

The main observations were as follows:(1)Breathing Modalities:-Approximately 14 (70%) patients preferred mouth breathing.-A total of 6 (30%) patients exhibited mixed respiratory patterns, using both the oral and nasal channels.-There was no evidence of a preferential nasal respiratory pattern in any of the cases.
(2)Respiratory Difficulties:-A total of 16 (80%) cases reported allergies (to pollen, dust, mites, etc.).-All subjects displayed nocturnal snoring, whereas 50% of the cases had obstructive apnea.
(3)Oral Posture at Rest:-In total, 12 (60%) patients demonstrated an open oral posture at rest.-Moreover, 8 (40%) patients showed a semi-closed oral posture at rest.-At rest, none of the cases presented a correct oral posture (a closed posture indicative of healthy respiratory mode).


Before the surgical intervention, closed nasality was also evaluated using the laryngeal mirror test, which revealed its presence in 80% of the examined patients. Similarly, nasal resonance was not observed in these cases.

Upon analyzing the vocal function of enrolled patients before surgery, it was observed that:-A total of 17 (85%) patients reported subjective symptoms closely related to phonatory activity.

Out of these 17 patients:
-A total of 14 (82.35%) patients demonstrated an actual change in their voice.-Although 3 (17.64%) patients complained of subjective phonatory activity disorders, they did not exhibit any vocal disorders.-The 3 patients who did not report subjective symptoms did not present dysphonia or vocal function changes.The subjective symptoms that patients reported most frequently were as follows:vocal fatigue: 62%;need to clear the throat: 50%; andoro-pharyngeal burning sensation: 75%

In addition to the subjective assessment of vocal function, an objective assessment using acoustic analysis was also performed, allowing us to define the psychoperceptual characteristics of the voice and the respiratory dynamics used in phonation. This analysis revealed the following:The pitch level of all individuals was found to be regular and appropriate for their gender and age.The vocal intensity was as follows:-normal in 8 (40%) cases;-high in 10 (50%) cases; and-inconsistent in 2 (10%) cases.
The vocal timbre was as follows:-personal (unique to the individual) in 4 (20%) cases;-hoarse in 13 (65%) cases; and-husky in 3 (15%) cases.The phonetic onset was as follows:-correct in 4 (20%) cases;-harsh in 13 (65%) cases; and-inconsistent in 3 (15%) cases.The vocal stability was as follows:-Regular in 8 (40%) cases;-Inconsistent in 12 (60%) cases.The duration of phonation was as follows:-regular in 8 (40%) cases; and-inconsistent (reduced) in 12 (60%) cases.The respiratory pattern during phonation was as follows:-chest-apical in 9 (45%) cases;-mixed in 11 (55%) cases; and-none of the patients exclusively demonstrated the costal-diaphragmatic respiratory mechanism.

Thus, the incidence of vocal disorders was particularly high, i.e., in 14 (70%) patients.

The phonetic inventory revealed that approximately 80% of the examined patients had articulation problems in their language, indicating that the incidence of articulation disorders in the pre-operative phase was quite high from a linguistic perspective. The following observations were made.

-Devoicing of the occlusive phonemes /b/, /d/, /g/, and sometimes also of the fricative phoneme /z/.-Interdental sigmatism, meaning the production of the fricative phoneme /s/ with the tip of the tongue placed between the upper and lower incisors.-Frequent interdentalizations of the alveolar phonemes /s/, /z/, /t/, /d/, /l/, /r/.-Rotacism, the production of the /r/ phoneme without the proper periodic vibration of the tip of the tongue.-Structure simplifications, i.e., modifications in the syllabic structure of words, such as reductions in consonant clusters, were also found in some patients.

Finally, the patients were also subjected to aerophonic examination, in which the phonatory flow indices (l/s) and subglottic pressure (cm H_2_O) were measured. The following observations were made in the pre-operative phase:-the phonatory flow was lower in 16 (80%) patients, while its higher values were observed in the remaining four (20%) patients; and-the subglottic pressure was higher in 17 (85%) patients, while it was lower in the remaining three (15%) patients than in the rest of the examined group.

### Normality Test

The Shapiro–Wilk test of normality rejects the null hypothesis of normal distribution (*p*-value < 0.001) for the parameters of age, numerical speech flow, and numerical subglottic pressure in the complete sample (Table 2). Repeating the subgroup analysis, we observe that age is significantly different (*p*-value < 0.05) from a normal distribution in all three groups (Table 3). The phonatory flow appears to have a normal distribution only in group A-post-surgery and the subglottic pressure only in Group B. Given that the hypothesis of normality must be verified in each subgroup when a parametric comparison test is made and that the sample size is not high enough, the comparison between these parameters in the groups will be done through non-parametric tests.

The treatment group (Group A-pre surgery) is compared with the control group (Group B) to verify the homogeneity of the following parameters: age, sex, respiratory pattern, oral posture at rest, subjective symptomatology, vocal intensity, vocal timbre, phonic attack, interdentalisation, phonatory flow and subglottic pressure (Table 4).

For each parameter the test (Chi-square test or the U Mann–Whitney) is not statistically significant (*p*-value > 0.05), therefore there is no evidence of significant difference or association between any of the variables analyzed between the treatment group and the control group.

We proceed to investigate whether the parameters of interest in group A post-surgery are distributed differently compared to the pre-surgery group (Table 5). All the Chi-squared tests are statistically significant (*p*-value < 0.05) therefore for each qualitative parameter we can state that there is an association with the variable that distinguishes the two groups. For the quantitative variables phonatory flow and subglottic pressure, the test for paired samples of Wilcoxon is significant (*p*-value < 0.001) i.e., there is a significant difference in the distributions of these two parameters between the pre- and post-surgery (Figure 1 and Figure 2).

## 4. Discussion

The results of our study allowed us to make several observations regarding the primary effects of adenotonsillectomy on vocal function, language articulation, and breathing. The results showed that these factors were significantly affected by the surgical removal of adenoids and tonsils. As R. Mora et al. noted in a 2007 study, the vocal tract, which extends from the glottis to the oral cavity, is an important resonator when we speak, and changes to this structure result in variations in language characteristics and voice [11,12,22]. Our study also found that approximately 75.5% (*n* = 15) of adenotonsillar hypertrophy patients had problems with vocal function, language, and breathing. This prevalence is consistent with that found in previous scientific literature [13,14,15,16]. Adenoid hypertrophy is responsible for sleep apnea in pediatric cases [17]. Adenotonsillectomy changes the structure of the vocal tract, which affects the resonance in the production of language and voice even though it does not directly affect the functionality of the larynx. Hypertrophic adenoids and tonsils can modify the dimensions of the pharyngeal orifice and primarily interfere with breathing. They create an obstruction that forces the subject to use the oral breathing route and causes them to hold their mouths at an incorrect oral posture at rest [13,16,17,18,19,20]. According to our data, 70% (*n* = 14) of the patients were mouth breathers with incorrect oral postures before the intervention. This condition affects the growth of the skull and facial structure, mandible, resting tongue posture, overall posture, and even the laryngeal muscles, which can result in dysphonia [1,2,3,21]. All these points supported and validated that 85% (*n* = 17) of patients experienced significant subjective symptoms during a phonatory activity during the pre-operative phase. The most frequently reported symptoms included vocal fatigue, a burning sensation, muscle pain, and needing to clear the throat. These findings were consistent with those reported in the literature by B. Jankowska et al. [10,11,12,23,24]. Contrary to what I. Lundeborg et al. reported, a visual analog scale (VAS), which included eight voice quality parameters, was used for perceptual analysis [18,19,20]. In our study, we examined the psycho-perceptive characteristics of the voice as expressed in the phonetic framework, including pitch, which is the psycho-perceptual sensation of the voice; intensity, which denotes the volume of the voice and refers to subglottic pressure; phonetic attack, which is the beginning of phonation, i.e., the moment when the vocal cords adduct; timbre, which defines the quality of the voice and is dependent on the dimensional relationships of the supra-glottic resonance cavities; phonetic hold, which is the ability to maintain all vocal parameters constant over time; and duration, which is the ability to maintain the correct glottal configuration for a certain period time. Our analysis showed that intensity, timbre, and attack were the parameters most affected by adenotonsillar hypertrophy. In comparison to normal, 50% (*n* = 10) of patients had increased intensity, 80% (*n* = 16) had a hoarse timbre, and 80% (*n* = 16) had a harsh attack. It was evident that infantile mouth breathing can cause the larynx to stretch upwards and downwards, resulting in the strain at the glottic level and possible dysphonia [1,2,3]. In accordance with previously published reports, the observed values of perturbation in the various parameters of voice decreased and returned to normal levels in the subsequent post-operative phase (e.g., E. Hultcrantz et al.) [18,19,20,21,22,23,24]. In 80% (*n* = 16) of cases before surgery, our data revealed a pattern of alteration in the pre-operative phase, also seen in the degree of nasality. However, this finding contradicts the findings of some studies, [15,16,17,18,22,23,24] as some reported normalization of resonance after surgery, while others found a normal pre-operative situation that became hypernasal post-operatively. This was also highlighted in our study, in which only two of the examined patients presented excessive nasality.

It was evident that hypertrophic tonsils and adenoids interfere with the transmission of sounds produced by the tongue in the oral cavity, such as s, r, l, d, and t, whose articulation point involves the tip of the tongue. In accordance with the findings of Salami et al., they may also reduce the number of vocal cord vibrations that determine voiced phonemes such as b, d, g, and z due to incomplete glottic closure [11,12,13,25]. Indeed, our data demonstrated that 80% (*n* = 16) of the participants exhibited devoicing of occlusive phonemes, whereas 65% (*n* = 13) displayed interdentalization of apico-alveolar phonemes.

Furthermore, our investigation aimed to assess whether adenotonsillectomy affects vocal function via aerophonic examination. During the initial pre-operative phase, 85% of the participants showed elevated subglottic pressure values, which correlated with high vocal intensity, whereas phonatory airflow values were lower. Thus, phonatory resistance, which represents the ratio between subglottic pressure (Psub) and phonatory airflow resistance (PFR), exhibited high values during the pre-operative phase. This situation changed after the adenotonsillectomy procedure, which removed the obstruction of hypertrophic adenoids and tonsils. Therefore, we observed higher phonatory airflow values and lower subglottic pressure values at the base, resulting in significantly lower phonatory resistance values post-surgery.

Our data analysis confirmed that hypertrophy of adenoids and tonsils significantly impacted breathing, vocal function, and speech articulation. They obstruct proper nasal breathing, essential for maintaining correct resting lip and tongue posture, resulting in mouth breathing and incorrect jaw and cervical posture changes [1,2,3,4,21,22,23,24,25,26].

These factors result in muscle tension in the neck region, which can also affect the vocal tract and lead to dysphonia [1,2,3,25]. Our study objectively observed this condition in assessing both the psycho-perceptive parameters of the voice and the indices of phonatory airflow and subglottic pressure recorded by the Articulatory Test of the aerophonic examination. These indices clearly showed that, in the pre-operative phase, adenotonsillar hypertrophy alters air passage through the vocal tract, increasing phonatory resistance and glottic effort. It was also observed that changes in the articulation of various phonemes affect language. All of this was observed before the surgical intervention, so a comparison could be made three months following adenotonsillectomy to determine its effect on these alterations. Our data confirmed that adenotonsillectomy restores balance in cases with grade 3–4 hypertrophy, as normal nasal breathing is re-established after the intervention. According to our data, the vocal function improved following the intervention, as most vocal parameters returned to normal and the discomfort associated with phonatory activity disappeared. Three months after the intervention, the aerodynamic indices of pressure and flow decreased and increased, respectively, resulting in a decrease in phonatory resistance. Even the articulation of language significantly improved.

Surely some limitations of the study are: small groups, different age, different grade of adenotonsillar hypertrophy, informed consent was signed by the parents.

## 5. Conclusions

In conclusion, this study highlighted the relationship between adenotonsillectomy and vocal function despite the need for further validation with a larger sample size. This was confirmed by statistically significant improvement of all study parameters after surgery. However, such benefits were not statistically evident in the group of non-operated patients.

In conclusion, adenotonsillectomy can improve a patient’s clinical status due to the resolution of recurrent inflammatory infections, as well as improvement in vocal characteristics and language acquisition, with the recovery of adequate nasal breathing and correction of altered postural attitudes.

## Figures and Tables

**Figure 1 jpm-13-01002-f001:**
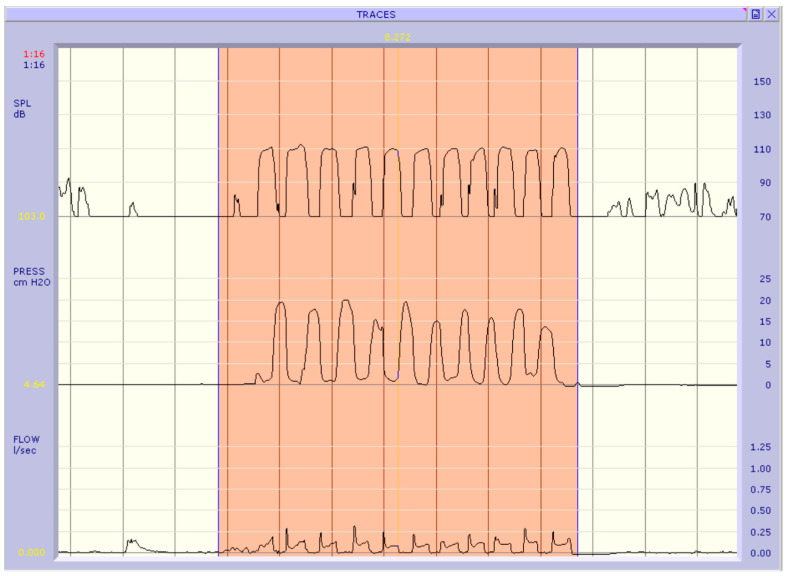
Aerophonic examination of a subject in the pre-operative phase: PFR: 0.097 l/s; Pk Psub: 16. 95 cm H_2_O.

**Figure 2 jpm-13-01002-f002:**
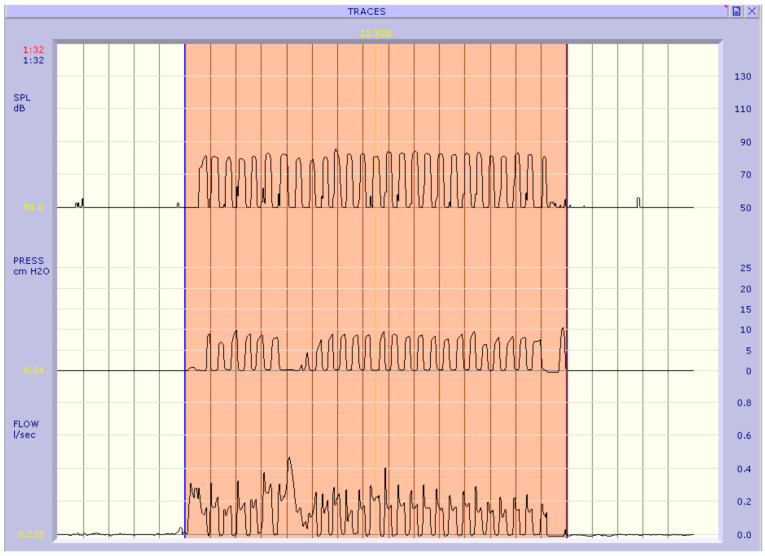
Aerophonic examination of a subject in the post-operative phase: PFR: 0.169 l/s; Pk Psub: 8.43 cm H_2_O.

**Table 1 jpm-13-01002-t001:** Clinical-demographic characteristics of the study participants of all 20 patients in group A.

		Mean	Standard Deviation	Count	Column N %
**Age**		6.75	2.53		
**Pharyngo-tonsillitis (Episodes per year)**		6.75	1.52		
**Sex**	F			10	50%
	M			10	50%
**Breathing Modalities**	Mixed			6	30%
	Mouth breathing			14	70%
**Allergies**	No			4	20%
	Yes			16	80%
**Sleep Disorders**	nocturnal snoring,			10	50%
	nocturnal snoring and obstructive apnea			10	50%
**Oral Posture at Rest**	Open			12	60%
	Semi-closed			8	40%
**Subjective symptoms**	No			3	15%
	Yes			17	85%
**Dysphonia**	No			6	30%
	Yes			14	70%
**Vocal intensity**	High			10	50%
	Inconsistent			2	10%
	Normal			8	40%
**Vocal timbre**	Personal			4	20%
	Hoarse			13	65%
	Husky			3	15%
**Phonetic onset**	Correct			4	20%
	Harsh			13	65%
	Inconsistent			3	15%
**Vocal stability**	Inconsistent			12	60%
	Regular			8	40%
**Duration of phonation**	Inconsistent			12	60%
	Regular			8	40%
**Respiratory pattern**	Mixed			11	55%
	Chest—Apical			9	45%
**Vocal Examination**	Negative			6	30%
	Positive			14	70%
**Speech articulation problems**	No			4	20%
	Yes			16	80%
**Disarticulation**	No			4	20%
	Yes			16	80%
**Sigmacism**	No			7	35%
	Yes			13	65%
**Rotacism**	No			11	55%
	Yes			9	45%
**Interdentalizations**	No			7	35%
	Yes			13	65%
**Reductions in consonant clusters**	No			8	40%
	Yes			12	60%
**Fonatory Flow Rate**	Higl			4	20%
	Low			16	80%
**Subglottal Pressure**	High			17	85%
	Low			3	15%
**Numerical Fonatory Flow**		0.10	0.03		
**Numerical Subglottal Flow**		14.95	2.53		

**Table 2 jpm-13-01002-t002:** Shapiro–Wilk test for the parameters age, numerical fonator flow, numerical subglottcal flow.

Shapiro-Wilk	Statistic	df	Sig.
**Age**	0.874	60	<0.001
**Numerical Fonatory Flow**	0.891	60	<0.001
**Numerical Subglottal Flow**	0.920	60	<0.001

**Table 3 jpm-13-01002-t003:** Shapiro–Wilk test in subgroup A (pre and post- surgery) and B.

		Shapiro-Wilk		
		Statistic	df	Sig.
GROUP A—POST SURGERY	**Age**	0.882	20	0.019
	**Numerical Fonatory Flow**	0.960	20	0.546
	**Numerical Subglottal Flow**	0.878	20	0.017
GROUP A—PRE SURGERY	**Age**	0.882	20	0.019
	**Numerical Fonatory Flow**	0.729	20	<0.001
	**Numerical Subglottal Flow**	0.895	20	0.033
GROUP B	**Age**	0.878	20	0.016
	**Numerical Fonatory Flow**	0.753	20	<0.001
	**Numerical Subglottal Flow**	0.914	20	0.076

**Table 4 jpm-13-01002-t004:** Comparison between Group-A (Pre-surgery) and Group—B.

		Group A—Pre Surgery			Group B			
		Median	Count	Column N %	Median	Count	Column N %	*p*-Value
**Age**		6.50			6.00			0.758 ^b^
**Sex**	F		10.00	50.00		10.00	50.00	1.000 ^a^
	M		10.00	50.00		10.00	50.00	
**Breathing Modalities**	Mixed		6.00	30.00		9.00	45.00	0.327 ^a^
	Mouth breathing		14.00	70.00		11.00	55.00	
**Oral Posture at Rest**	Open		12.00	60.00		11.00	55.00	0.749 ^a^
	Semi-closed		8.00	40.00		9.00	45.00	
**Subjective symptoms**	No		3.00	15.00		5.00	25.00	0.429 ^a^
	Yes		17.00	85.00		15.00	75.00	
**Vocal intensity**	High		10.00	50.00		10.00	50.00	0.372 ^a^
	Inconsistent		2.00	10.00		5.00	25.00	
	Normal		8.00	40.00		5.00	25.00	
**Vocal timbre**	Personal		4.00	20.00		9.00	45.00	0.082 ^a^
	Hoarse		13.00	65.00		6.00	30.00	
	Husky		3.00	15.00		5.00	25.00	
**Phonetic onset**	Correct		4.00	20.00		5.00	25.00	0.234 ^a^
	Harsh		13.00	65.00		8.00	40.00	
	Inconsistent		3.00	15.00		7.00	35.00	
**Interdentalizations**	No		7.00	35.00		5.00	25.00	0.490 ^a^
	Yes		13.00	65.00		15.00	75.00	
**Fonatory Flow Rate**		0.09			0.08			0.253 ^b^
**Subglottal Pressure**		15.77			16.07			0.678 ^b^

^a^ Chi-square test, ^b^: U Mann–Whitney.

**Table 5 jpm-13-01002-t005:** Comparison between Group A pre- and post-surgery.

		Group A—Post-Surgery			Group A—Pre-Surgery			
		Median	Count	Column N %	Median	Count	Column N %	*p*-Value
**Breathing Modalities**	Mixed		2.00	10.00		6.00	30.00	<0.001 ^a^
	Nasal breathing		18.00	90.00		0.00	0.00	
	Mouth breathing		0.00	0.00		14.00	70.00	
**Oral Posture at Rest**	Open		0.00	0.00		12.00	60.00	
	Close		18.00	90.00		0.00	0.00	<0.001 ^a^
	Oral		2.00	10.00		0.00	0.00	
	Semi-closed		0.00	0.00		8.00	40.00	
**Subjective symptoms**	No		18.00	90.00		3.00	15.00	<0.001 ^a^
	Yes		0.00	0.00		17.00	85.00	
**Vocal intensity**	High		3.00	15.00		10.00	50.00	0.011 ^a^
	Inconsistent		0.00	0.00		2.00	10.00	
	Normal		17.00	85.00		8.00	40.00	
**Vocal timbre**	Personal		15.00	75.00		4.00	20.00	0.002 ^a^
	Hoarse		5.00	25.00		13.00	65.00	
	Husky		.00	0.00		3.00	15.00	
**Phonetic onset**	Correct		16.00	80.00		4.00	20.00	<0.001 ^a^
	Harsh		4.00	20.00		13.00	65.00	
	Inconsistent		0.00	0.00		3.00	15.00	
**Interdentalizations**	No		20.00	100.00		7.00	35.00	<0.001 ^a^
	Yes		0.00	0.00		13.00	65.00	
**Fonatory Flow Rate**		0.15			0.09			<0.01 ^b^
**Subglottal Pressure**		11.00			15.77			<0.001 ^b^

^a^ Chi-square test, ^b^:U Mann–Whitney.

## Data Availability

Data sharing not applicable; No new data were created or analyzed in this study. Data sharing is not applicable to this article.

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
