# Peer review of "Effects of Adenotonsillectomy on Vocal Function"

_jpm, 2023, doi:10.3390/jpm13061002_

Round 1

Reviewer 1 Report

Please, find the additional remarks:

1. There is a need to present some factual information in the abstract (number of children enrolled, their age, etc.)

2. Statistical analysis. Please, indicate how the normality of data distribution was tested prior to statistical analysis.

3. Results. Please, add table 1 with clinical-demographic characteristics of the study participants comparing the study and control groups and presenting the resulting test of difference.

4. Results. Please, combine Tables 1-7 in one table (Table 2) comparing the study and control groups and presenting the resulting test of difference.

5. Results. For Tables 8-9 add the number and percentage of patients pre- and post-op (Table 8) and the number and percentage of patients in the study and control groups (Table 9). In addition, indicate what test of difference was used in each particular Table. Since Table 8 presents the data for the same group over a period of time, Mann-Whitney test is not appropriate. Please, test each variable for the normality of data distribution and use either Student's t-criterion for paired samples (if the data distribution is normal) OR Wilcoxon test (if the data distribution is different from normal)

6. Discussion. Please, describe limitations of your study

7. Conclusion. Please, shorten this section by moving some of the information to Discussion.

Author Response

REVISORE 1

Please, find the additional remarks:

  1. There is a need to present some factual information in the abstract (number of children enrolled, their age, etc.)

- I added what was requested in the abstract

  1. Statistical analysis. Please, indicate how the normality of data distribution was tested prior to statistical analysis.

- I’ve added LINE 160

  1. Results. Please, add table 1 with clinical-demographic characteristics of the study participants comparing the study and control groups and presenting the resulting test of difference.

- I’ve added

  1. Results. Please, combine Tables 1-7 in one table (Table 2) comparing the study and control groups and presenting the resulting test of difference.

- I’ve added unique table (table 1)

  1. Results. For Tables 8-9 add the number and percentage of patients pre- and post-op (Table 8) and the number and percentage of patients in the study and control groups (Table 9). In addition, indicate what test of difference was used in each particular Table. Since Table 8 presents the data for the same group over a period of time, Mann-Whitney test is not appropriate. Please, test each variable for the normality of data distribution and use either Student's t-criterion for paired samples (if the data distribution is normal) OR Wilcoxon test (if the data distribution is different from normal)

- Correct

  1. Discussion. Please, describe limitations of your study

- ok

  1. Conclusion. Please, shorten this section by moving some of the information to Discussion.

I corrected what you requested

Reviewer 2 Report

Esteemed colleagues,

You present a very interesting case regarding the improvement of voice quality after tonsillectomy procedures.

However, there are some aspects that require your attention.

I believe that there is a redundancy between the text and the tables you should remove the tables and maintain the data description only in the text, because all your tables have around 5 lines you can easily enumerate all the categories in the text.

Moreover, given the small scale of the study group you could include a table in landscape mode to present all the characteristics of the study group subjects.

In the discussion group include a small section mentioning the limitations of the study: small group, informed consent from the parents, etc.

Furthermore, in the discussion section underline the impact on sleep apnea and reference this to newer articles from MDPI platform such as Neagos A, Dumitru M, Neagos CM, Mitroi M, Vrinceanu D. Correlations between Morphology, the Functional Properties of Upper Airways, and the Severity of Sleep Apnea. J Clin Med. 2022 Sep 12;11(18):5347. doi: 10.3390/jcm11185347. PMID: 36142994; PMCID: PMC9502432.

Also correct the funding section at the end of the manuscript.

Appendix A is not available in the platform, please include the appendix as a supplementary material.

In the conclusion mention the necessity of a future study on a bigger number of patients.

Looking forward to receiving your improved manuscript.

Author Response

Esteemed colleagues,

You present a very interesting case regarding the improvement of voice quality after tonsillectomy procedures.

However, there are some aspects that require your attention.

I believe that there is a redundancy between the text and the tables you should remove the tables and maintain the data description only in the text, because all your tables have around 5 lines you can easily enumerate all the categories in the text.

  • ok

Moreover, given the small scale of the study group you could include a table in landscape mode to present all the characteristics of the study group subjects.

 -ok

In the discussion group include a small section mentioning the limitations of the study: small group, informed consent from the parents, etc.

  • ok

Furthermore, in the discussion section underline the impact on sleep apnea and reference this to newer articles from MDPI platform such as Neagos A, Dumitru M, Neagos CM, Mitroi M, Vrinceanu D. Correlations between Morphology, the Functional Properties of Upper Airways, and the Severity of Sleep Apnea. J Clin Med. 2022 Sep 12;11(18):5347. doi: 10.3390/jcm11185347. PMID: 36142994; PMCID: PMC9502432.

  • correct

Also correct the funding section at the end of the manuscript.

  • correct

Appendix A is not available in the platform, please include the appendix as a supplementary material.

  • Correct

In the conclusion mention the necessity of a future study on a bigger number of patients.

  • correct

Looking forward to receiving your improved manuscript.

Round 2

Reviewer 1 Report

Dear authors,

My congratulations, you've done excellent job!

Please attach Tables 2 and 3 as a supplementary file. There is no need to include the Shapiro-Wilk test statistics in the results. Additionally, kindly remove the Italian terms from Table 3 (Gruppo B, etc.).

There are minor linguistic inconsistencies, and having the paper edited by a native speaker would be beneficial.

Author Response

Dear authors,

My congratulations, you've done excellent job!

Please attach Tables 2 and 3 as a supplementary file. There is no need to include the Shapiro-Wilk test statistics in the results. Additionally, kindly remove the Italian terms from Table 3 (Gruppo B, etc.).

There are minor linguistic inconsistencies, and having the paper edited by a native speaker would be beneficial.

Dear Reviewer,

i made the changes to english in the tables. Tables 1 and 2 have already been sent as supplementary material. I leave them in the text so that the editor can possibly correct them in the final draft.

We prefer to leave the Shapiro-Wilk test for clarity.

Thank you for taking the time to make corrections.